# Characteristics of Giant Cell Tumor of the Bone in Pediatric Patients: Our 18-Year, Single-Center Experience

**DOI:** 10.3390/children8121157

**Published:** 2021-12-08

**Authors:** Woo-Jong Kim, Sungmin Kim, Dae-Woong Choi, Gil-Hwan Lim, Sung-Taek Jung

**Affiliations:** Department of Orthopedic Surgery, Chonnam National University Medical School and Hospital, Gwangju 61469, Korea; overnoveragain@naver.com (W.-J.K.); kimsum83@gmail.com (S.K.); brokenmouse@naver.com (D.-W.C.); ragtag31@gmail.com (G.-H.L.)

**Keywords:** giant cell tumor, pediatric patients

## Abstract

A giant cell tumor (GCT) of the bone is characteristically found in skeletally mature patients. The tumor is rare in pediatric patients, and incidence reported in literature varies from 1.8% to 10.6%. We performed a retrospective study addressing symptoms, treatment, and outcome in pediatric patients who were diagnosed with GCT between March 1997 and January 2015 at our hospital. Fourteen (11.1%) of 126 surgically treated patients with histologically proven GCT were <19 years of age. We confirmed skeletal maturity using magnetic resonance imaging (MRI). Fourteen patients from 8 to 19 years old were identified. Sixteen lesions (76.2%) were found in long bones and 5 lesions (23.8%) in short bones. The most common site was around the knee in 8 patients (38%). GCTs mostly occur at the epi-metaphysis in 11 patients (52.3%). Regardless of the openness of epiphyseal plate, we observed GCT of bone in the epiphysis. Further study will be needed to prove the association between the presence of epiphyseal plate and location of tumor. Three patients (21.4%) had multicentric lesions, and four patients (28.5%) had local recurrence. Multicentric giant cell tumor and local recurrence occur more often in pediatric patients. The characteristics of GCT in pediatric patients do not differ from what is reported for GCT in adults.

## 1. Introduction

Giant cell tumors (GCTs) of the bone account for 21% of all benign tumors and 5% of all primary bone tumors [1]. In the 2020 World Health Organization classification, GCT of the bone was redefined from a benign to intermediate group [2]. GCT of the bone is characteristically found in skeletally mature patients and is most prevalent during the third decade of life [1]. GCTs are intermediate, can be locally aggressive and metastasize to the lungs. GCTs often develop in the epiphysis of long bones, and approximately 50% of cases develop GCTs around the knees [1].

GCTs are exceptionally rare in children, but there have been few reports that have documented GCT in pediatric patients. The characteristics of GCT in pediatric patients, such as incidence, location, local recurrence, and multicentric lesions, vary widely [1,3,4,5,6]. The purpose of this study was to document the characteristics of GCT of the bone in pediatric patients and to study the course of the disease with respect to its adult counterpart.

## 2. Materials and Methods

Our institutional review board (IRB) approved this study. We retrospectively reviewed 126 patients who were treated for GCT of the bone between March 1997 and January 2015 at our hospital. All patients underwent surgery, and pathology teams evaluated the specimens. The inclusion criteria for the present study were as follows: (1) patients younger than 19 years; (2) histologically confirmed conventional GCT; and (3) no evidence of hyperparathyroidism.

Patients were excluded when (1) the duration of follow-up was <2 years, and (2) only a bone needle biopsy was performed.

The 20 patients were under 19 years of age. Six patients were excluded from the study for the following reasons: three patients, who were lost to follow-up and three patients who underwent bone-needle biopsy only, were excluded. A total 14 patients (9 boys and 5 girls) with GCT were included in this study.

The patients’ records were studied for clinical data, chief complaint at the time of outpatient visit, surgical record, follow-up records, duration of pain, presence of pathologic fracture, primary GCT or recurrence, pulmonary metastasis, multicentricity, operative methods, and complications.

We used chronological age because, in some cases, it was difficult to clearly establish whether the epiphyseal growth plate was closed, and the patient’s age was considered a more appropriate parameter for this study than skeletal maturity.

Radiographs of the primary lesions at initial presentation were analyzed to determine the extent of tumor involvement and skeletal immaturity. The age at presentation and opening of the epiphyseal plate defined skeletal immaturity. If skeletal immaturity was difficult to ascertain with targeted radiograph (Figure 1A), we confirmed it using magnetic resonance imaging (MRI) (Figure 1B).

Most of the lesions were treated with cement augmentation or bone graft after intralesional curettage, except those in the so-called expendable bones, such as the upper end of the fibula, which were excised. We routinely used a high-powered burr, which is an electrocautery system. We also used local adjuvants, such as alcohol and cryotherapy with liquid nitrogen to reduce local recurrence. All surgical treatments were performed by a single surgeon.

All slides and blocks of the GCTs excised from the patients were reviewed by a pathologist who specializes in musculoskeletal oncology to confirm the diagnosis, which was made according to standard histological criteria.

The postoperative surveillance schedule included visits every 3 months for 2 years and 6 months thereafter. Local radiographs and chest radiographs were obtained every 6 months. All patients had at least 2 years of follow-up.

## 3. Results

The mean age of the patients at surgery was 15.6 (8–19) years. In the 14 patients, there were 21 primary lesions, and the mean follow-up period was 8.0 (2.0–24.0) years. A summary of the clinical data of the 14 patients is presented in Table 1.

The most common location was around the knee joint (eight lesions, 38%) followed by the proximal humerus (four lesions, 19%) in our cohort. Other sites were the metatarsal bone (two lesions), intermediate cuneiform (two lesions), and proximal fibula, proximal femur, distal fibula, distal ulna, and phalanx of the finger (one lesion each) (Table 2).

Out of the 17 lesions in long tubular bones in our study, 11 (52.3%) involved the epi-metaphysis (Figure 2) and six (28.5%) involved the metaphysis. However, no lesion involved the epiphysis only, and no mid-diaphyseal localization was observed.

The growth plates of long tubular bone in 11 patients (78.5%) were open, whereas that of one patient (7.1%) was closed.

Radiologically, all the lesions were lytic in nature, with margins ranging from sclerotic to ill-defined. The original contour of the bone was frequently altered, usually with expansile remodeling. In most of the large, long bones, such as the tibia, humerus, and femur, the GCT was typically eccentric and limited solely to the metaphysis. In these locations, the bone contour was either normal or demonstrated mild expansile remodeling. Larger lesions tended to be more centrally located. In the slender and smaller long bones, such as the radius, fibula, and metacarpals, the GCT was almost invariably centrally located, filling the metaphysis and extending into the metadiaphysis and sometimes the diaphysis, often resulting in a significantly remodeled and expanded bone contour.

Multiple synchronous or metachronous lesions in single or multiple bones are defined as multicentric GCTs [7]. In our study, three patients (21.4%) had multicentric GCTs; two patients had GCTs around the knee and one patient had a GCT in the metatarsal. Additionally, one patient had both synchronous and metachronous lesions, and two patients had synchronous lesions.

Five local recurrences developed in 4 patients (28.5%). The average time to recurrence was 1.5 (0.5–2.7) years. At the time of recurrence, the patients were asymptomatic. Both multicentric and recurrent lesions occurred simultaneously in two out of the 14 patients (14.2%).

Plain chest radiographs or chest CT scans were obtained from the 14 patients prior to surgery, which confirmed that there were no cases of pulmonary metastasis before surgery. During follow-up, a patient had lung metastasis without local recurrence. This was managed by the cardiothoracic team, and the mass was removed by video-assisted thoracic surgery. Later, the lung mass recurred, and it was successfully removed using the previous method.

According to the Campanacci grading system [8] six (42.8%) were grade 3, seven were grade 2 and one (7.1%) was grade 1.

## 4. Discussion

GCT is a relatively rare and intermediate tumor in young adults, but it can be locally aggressive and may metastasize to the lungs. GCT cases in pediatric patients have not been well studied, especially in those aged <19 years who have immature bones, as it is not easy to clearly establish whether the growth plate is closed. Skeletal maturation depends on the age and sex of a person and varies between individuals. This is the reason why the inclusion of skeletally immature patients was diverse in previous studies. Hence, Picci et al. [4] restricted their study to patients under 15 years of age and in whom both sides of the epiphyseal plate had to be evident on plain radiographs or tomograms. Kransdorf et al. [5] reported on 50 skeletally immature patients from the 876 cases seen in the Armed Forces Institute of Pathology; they only included patients with a “radiographically open physis.” In our study, we reviewed the pathophysiological features of 14 out of 126 cases (11.1%) of GCT that were below 19 years old. In addition, we checked whether the physeal line was still open or closed in all cases using MRI.

Our study showed that the most common site of involvement was around the knee, with eight out of 21 lesions (38.0%) developing in the lower end of the femur and upper end of the tibia. Picci et al. [4] reported a series of six children with lesions around the knee joint. In their report, Puri et al. [6] revealed that 9 out of 17 lesions (53%) were around the knee joint. Other publications regarding children have reported that 22–53% of lesions developed around the knee (Table 3) [4,5].

In adults, the most frequent locations of GCT, in decreasing order, are the distal femur, proximal tibia, distal radius, and sacrum; half of the GCTs arise in the knee region [9,10,11]. As a result, the most common area of involvement of GCT lesions was around the knee in adults and children.

We reported that the metatarsal bones, cuneiform, and phalanx were rarely involved, being affected in only five out of 21 lesions (23.8%). In the small bones, the common areas of involvement were the hands and feet. After treatment, no local recurrence or complications were observed. In previous studies, small bone involvement was rare in GCTs, with 90% of GCTs being found in long bones [12,13]. Kransdorf et al. [5] found several (34%) lesions in the small bones of the hands and feet. The small bones of the hands and feet, vertebrae, pelvis, and skull are rarely involved [13]. In our study, GCT occurred mainly in the hands and feet, and there was no spinal involvement. None of the patients had GCT in lesions that were difficult to access surgically, such as the vertebral column or sacrum. The use of denosumab could be taken into consideration; however, we did not use it because sufficient resections were already performed in all cases. Besides, there were many concerns about adverse effects such as rebound bone turnover and hypercalcemia [14]. There is insufficient evidence to conclude that small bone involvement is more common in pediatric patients from our small case series alone. Further studies are recommended to properly determine small bone involvement.

Lesions involving the metaphysis or the diaphysis without epiphyseal involvement are exceptionally rare in adults. In our patients, six of 21 lesions in tubular bones were metaphyseal, and 11 out of 21 lesions in tubular bones were epi-metaphyseal at various ages. However, there were no cases of epiphyseal or diaphyseal lesions. Breaching of the epiphyseal plate and infiltration of the epiphysis is an uncommon feature of benign lesions. This locally aggressive behavior is not an uncommon feature of GCTs. Invasion of the epiphysis by the tumor is considered a sign of aggressive growth [3,15]. Picci et al. [4] reported that in five out of six patients in whom the epiphysis was penetrated by tumor tissue, the tumor had an aggressive tendency. In spite of its clinically aggressive behavior, the atypical histological patterns could not be identified. Further studies on the correlation between histopathological characteristics and the location of GCTs are needed.

The recurrence rate of GCTs in this study was 33% (4 of 12 patients), which was higher than that in other studies (Table 3). The local recurrence rate of GCT in adult solitary lesions was reported to range from 0% to 25% [16,17,18]. Hoch et al. [7] reported that the single most important factor related to the risk of recurrence was the incomplete removal of the lesion. The local recurrence rate was 37% in patients treated with intralesional curettage and 5% in those treated with wide excision. Furthermore, radical surgeries, such as en bloc resection, have been advocated to avoid local recurrence. The principles of management remain the same even in cases of recurrent tumors, and all local recurrences are managed with repeat curettage. In cases of tumor recurrence, tumor prosthesis or en bloc excision is considered. Vult von Steyern et al. [19] reported that local recurrence of GCT in long bones after treatment with curettage and cementing can generally be successfully treated with further curettage and cementing, with only a minor risk of increased morbidity.

GCTs typically present as solitary lytic lesions; however, in approximately 1% of cases [13,20,21,22], they manifest as multiple synchronous or metachronous lesions in single or multiple bones. Hoch et al. [7] reported that in patients with multicentric GCTs, 59% were younger than 20 years and were considerably younger than those with solitary GCTs. In addition, GCTs in skeletally immature patients had a higher incidence of local recurrence compared to those in skeletally mature patients. In our study, three of 14 patients (2.14%) had multicentric lesions, and four of 14 patients (28.5%) had a recurrence.

In this study, five of 14 (35.7%) patients with GCT were female. Some studies have suggested a slight female predilection for GCT [9,13,23], but others have found no difference between genders [1,12]. Due to the small number of subjects in this study, the results seem to contradict those of previous studies. Dahlin et al. found that 56% of adult patients with GCT were female and 72% were <20 years of age [22]. Interestingly, numerous publications have shown a female predominance (60–82%) in the younger age groups [3,4,6].

Children differ from adults in terms of residual growth, and deformity and growth failure may occur due to the occurrence of GCT. In our study, GCT occurred at the distal phalanx in an 8-year-old male patient, which was resected immediately. Thereafter, no deformity occurred until the last follow-up after 2 years. Most of the patients did not have much residual growth due to old age at the time of diagnosis, and no other deformity had occurred in our cases after treatment. Further studies on the correlation between residual growth and deformity are needed.

Modern experimentations such as proteomic screening can help us identify the metastatic potential and elucidate the mechanism of GCT pathology. Certain factors in primary tumors such as glutathione peroxidase 1 are strongly related to metastasis [24]. Ambrosi et al. [25], a driver mutation in the histone 3.3 (H3.3) gene H3F3A has been identified in as many as 96% of GCTs of bone. Immunohistochemical and molecular detection of H3F3A gene mutation represents a reliable diagnostic tool to distinguish from its mimickers due to the difference in prognosis and treatment. The use of cutting-edge scientific methods to uncover the pathophysiological scheme of GCT will be essential in the discovery of new and more effective treatment options [26].

There are several limitations to our study. First, a relatively small number of patients (14 patients) were included because of the low incidence rate of GCT. Second, although partial openness of the epiphyseal plate was also checked as the age increased, we equated the degree of partial epiphyseal plate openness with full opening. Therefore, in this study, we recruited patients according to their age rather than skeletal maturity. Lastly, we wanted to investigate the pathological grade according to the extent of metaphysis and epiphysis involvement, but there are limitations to comparing pathologic data, as in the previous study.

## 5. Conclusions

GCTs of the bone usually occur in adults, but they can also rarely develop in pediatric patients whose skeletons are immature. In our institution, GCTs of the bone developed in 14 patients who were <19 years of age. There was no significant difference between the characteristics of GCT in adults and children. GCTs mostly occur at the epi-metaphysis, and the rates of multicentric lesions and local recurrence were higher than in adults.

## Figures and Tables

**Figure 1 children-08-01157-f001:**
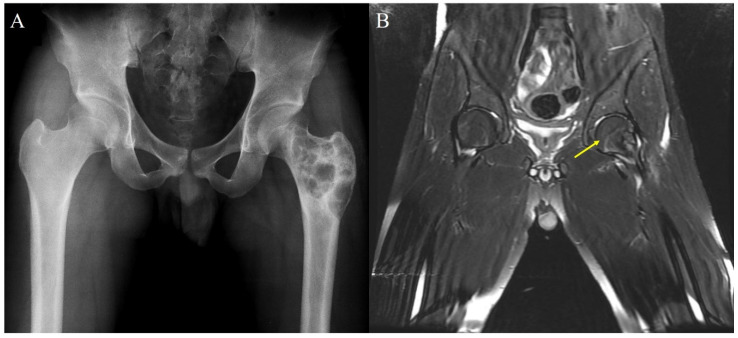
(**A**) Typical example of skeletal immaturity difficult to ascertain with targeted radiograph in 18-year-old male. (**B**) Magnetic resonance imaging (MRI) showing an open epiphyseal plate (arrow).

**Figure 2 children-08-01157-f002:**
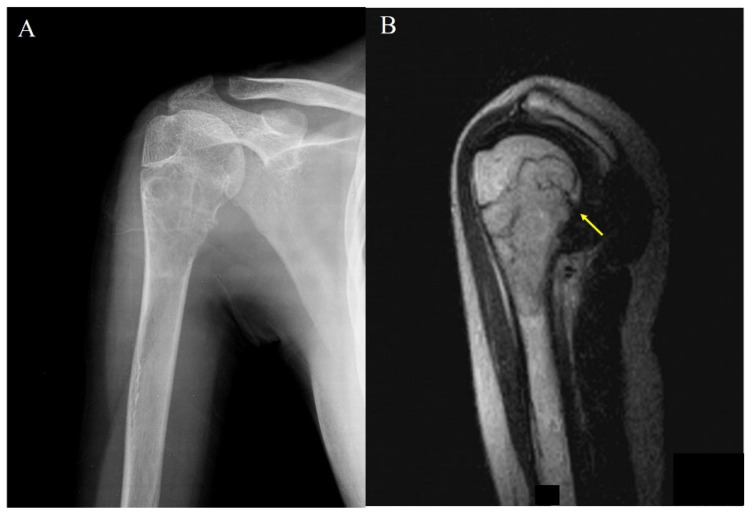
(**A**) Plain radiograph shows an osteolytic lesion of proximal humerus in a 15-year-old female. (**B**) Magnetic resonance imaging (MRI) showing an open epiphyseal plate and penetration of the tumor (arrow) into the epiphysis.

**Table 1 children-08-01157-t001:** Patient characteristics.

Case	Age (y)	Sex	Bone	Region	P/R	Grade	Surgery	Follow-Up (mo)	Remarks
1	13	F	Prox Humerus	M	P	3	Curettage + auto B/G	49	Incidentally detect, fracture history, LR 14 mo
1	17	F	Distal femur + Prox Tibia	EM (Femur) + M (Femur, Tibia)	P	2	Curettage + Excision (LR) + Prosthesis (LR)	239	LR 11 mo
2	18	M	Prox Humerus	EM	P	2	Curettage	101	5 months pain
3	19	M	Distal Ulnar	EM	P	2	Curettage + Excision + allo B/G	48	1 month pain, LR 2 times (6, 8 mo)
4	19	M	Intermediate Cuneiform	-	P	2	Curettage + allo B/G	31	1 year pain
5	15	F	4,5 Metatarsal	-	P	2	Curettage	256	3 months pain
6	14	M	Distal femur + Prox Tibia (Rt) Tibia + Prox Fibular (Lt)	EM (Femur, Tibia, Fibular)	P	1	Curettage + Excision (LR) + Prosthesis (LR)	57	1 week pain, LR 28 mo
7	15	F	Prox Humerus	EM	P	3	Curettage	141	Incidentally detect, pathologic fracture
8	18	M	Prox Femur	M	P	3	Curettage	114	Incidentally detect, 2 lung metastasis
9	16	F	Dist Radius	EM	P	3	Curettage	85	2 weeks pain, pathologic fracture, LR 23 mo
10	12	M	Prox Tibia	M	P	2	Curettage	69	10 months pain
11	16	F	Dist Femur	EM	P	2	Curettage	41	1 month pain
12	19	M	Dist Fibular	EM	P	3	Curettage	53	6 months pain
13	18	M	Prox Humerus	M	P	1	Curettage	36	1 year pain
14	8	M	Finger Phalanx	-	P	1	Curettage + allo B/G	24	1 month pain

EM: Epi-metaphyseal M: Metaphyseal P: Primary R: Recurrent C: Concentric E: Eccentric LR: Local recurrence B/G: Bone graft Eleven of the 14 (71.4%) patients had pain, whereas GCT was detected incidentally in three patients. Pathologic fractures were found in two patients. The duration of symptoms prior to surgery averaged 0.4 (0.04–1) years.

**Table 2 children-08-01157-t002:** Radiographic assessment of giant cell tumor in pediatric patients.

	Total of 14 Patients, 21 Lesions
Anatomical site (Lesions)	
Adjacent to knee joint	8 (38.0%)
Distal femur	4 (19.0%)
Proximal tibia	4 (19.0%)
Proximal fibula	1 (4.7%)
Proximal humerus	4 (19.0%)
Metatarsal bone	2 (9.5%)
Proximal femur	1 (4.7%)
Distal fibula	1 (4.7%)
Distal ulna	1 (4.7%)
Intermediate cuneiform	2 (9.5%)
Phalanx (finger)	1 (4.7%)
Physeal closure (Patients)	
Open physis	11 (78.5%)
Closed physis	1 (7.1%)
N/A (d/t short bone)	2 (14.2%)
Location (Lesions)	
Physeal involvement	
Metaphysis	6 (28.5%)
Epi-metaphysis	11 (52.3%)
Epiphysis	4 (19.0%)
N/A (d/t short bone)	2 (14.2%)

N/A: Not available.

**Table 3 children-08-01157-t003:** Research data from other studies about children and adolescents with GCT of the bone.

	Incidence	Primary Patient Number	Most Common Site	Most Common Location	Recurrence
Picci et al. [3]	1.7%	6	Distal femur (4 lesions, 67%)	Epi-metaphysis (5 lesions, 83%)	N/A
Puri et al. [5]	6%	11	Distal femur (5 lesions, 29%)	Epi-metaphysis (13 lesions, 76%)	20%
Kransdorf et al. [4]	5.7%	50	Proximal tibia (9 lesions, 18%)	Metaphysis (48 lesions, 96%)	N/A
Schutte et al. [2]	10.6%	49	Proximal tibia (10 lesions, 20%)	Epi-metaphysis (27 lesions, 75%)	8%

N/A: not available; GCT: giant cell tumor.

## Data Availability

Not applicable.

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
