# Peer review of "Characteristics of Giant Cell Tumor of the Bone in Pediatric Patients: Our 18-Year, Single-Center Experience"

_children, 2021, doi:10.3390/children8121157_

Round 1

Reviewer 1 Report

Thank you for giving me an opportunity to review your manuscript. This manuscript is interesting.

The authors should write more about deformity and growth failure due to growth plate disorders following surgery of GCT in children. Did the authors treat the growth plate with alcohol or cryotherapy?

Line 27,29,124: According to WHO classification 2020, the GCTB has changed from benign to intermediate. Please correct.

Method: What the authors fill after curettage? Bone cement or allograft?

Table 1: Why does Table 1 include patients with a follow-up period of less than 2 years though the authors excluded patients with a follow-up period of less than 2 years?

Result: Did any patients receive denosumab or zoledronic acid?

Please read and cite the following paper because it includes important information.

Giant Cell Tumor of Bone in Patients under 16 Years Old: A Single-Institution Case Series Cancers 2021, 13(11), 2585; https://doi.org/10.3390/cancers13112585 - 25 May 2021

Author Response

Response to Reviewer 1 Comments

Thank you for giving me an opportunity to review your manuscript. This manuscript is interesting.

Point 1: The authors should write more about deformity and growth failure due to growth plate disorders following surgery of GCT in children.

Response 1: We agree with your suggestion. Children differ from adults in terms of residual growth, and deformity and growth failure may occur due to the occurrence of GCT. In our study, most of the patients did not have much residual growth due to old age at the time of diagnosis, and no other deformity had occurred in our cases after treatment. We have added the information regarding deformity and growth failure. We have described it in the lines 195–200.

Lines 195-200: Children differ from adults in terms of residual growth, and deformity and growth failure may occur due to the occurrence of GCT. In our study, GCT occurred at the distal phalanx in an 8-year-old male patient, which was resected immediately. Thereafter, no deformity occurred until the last follow-up after two years. Most of the patients did not have much residual growth due to old age at the time of diagnosis, and no other deformity had occurred in our cases after treatment. Further studies on the correlation between residual growth and deformity are needed.

Point 1-1:

Did the authors treat the growth plate with alcohol or cryotherapy?

Response 1-1: Thank you for your comments. We used local adjuvants, such as alcohol and cryotherapy with liquid nitrogen to reduce local recurrence. Regardless of the location of the tumor (in metaphysis or epiphysis), the same local adjuvants were applied with or without a growth plate involvement. Because there was not much residual growth due to old age at the time of diagnosis, no other deformity had occurred in our cases after treatment.

Point 2: Line 27,29,124: According to WHO classification 2020, the GCTB has changed from benign to intermediate. Please correct.

Response 2: Thank you for your comments. As you pointed out, in the WHO classification 2020, the GCTB was changed from benign to intermediate group. We have modified the group of GCT from benign to intermediate in lines 28,29 and 123.

Lines 28-29: In the 2020 World Health Organization classification, GCT of the bone was re-defined from benign to intermediate group(2).  

Point 3: Method: What the authors fill after curettage? Bone cement or allograft?

Response 3: Thank you for your comments. Most of the lesions were treated with cement augmentation or bone graft after intralesional curettage. Four patients had bone graft after curettage. We have modified the Lines 60-61 and Table 1. We have also added information regarding the bone graft in Table 1.

Lines 60-61: Most of the lesions were treated with cement augmentation or bone graft after intralesional curettage.

Point 4: Table 1: Why does Table 1 include patients with a follow-up period of less than 2 years though the authors excluded patients with a follow-up period of less than 2 years?

Response 4: Thank you for your comments. In Table 1, all the patients whose follow-up period was less than two years were considered as cases of local recurrence. In all cases, from the first surgery to the final follow-up, the duration of follow-up was more than two years. We have modified the local recurrence notation in Table 1.

Point 5: Result: Did any patients receive denosumab or zoledronic acid?

Response 5: Thank you for your comments. We did not use denosumab or zoledronic acid for treatment of patients. The reason for exclusion of denosumab in the treatment is added in lines 156-158.

Lines 156-158: The use of denosumab could be taken into consideration; however, we did not use it because sufficient resections were already performed in all cases. Besides, there were many concerns about adverse effects such as rebound bone turnover and hypercalcemia(14).

Point 6: Please read and cite the following paper because it includes important information.

Giant Cell Tumor of Bone in Patients under 16 Years Old: A Single-Institution Case Series Cancers 2021, 13(11), 2585; https://doi.org/10.3390/cancers13112585 - 25 May 2021

Response 6: Thank you for your comments. The summary of this paper was the application of histone 3.3 (H3.3) gene mutations as a diagnostic tool in the differential diagnosis of giant cell tumor of bone with its mimickers. We have added some information for modern experimentations of GCT based on this article. We have described it in lines 201-208.

Lines 201-208: Modern experimentations such as proteomic screening can help us identify the metastatic potential and elucidate the mechanism of GCT pathology. Certain factors in primary tumors such as glutathione peroxidase 1 are strongly related to metastasis(24). Ambrosi et al.(25), a driver mutation in the histone 3.3 (H3.3) gene H3F3A has been identified in as many as 96% of GCTs of bone. Immunohistochemical and molecular detection of H3F3A gene mutation represents a reliable diagnostic tool to distinguish from its mimickers due to the difference in prognosis and treatment. The use of cutting-edge scientific methods to uncover the pathophysiological scheme of GCT will be essential in the discovery of new and more effective treatment options.

Reviewer 2 Report

This is a well-written paper covering a very small patient population. Do the authors have any experience with the use of denosumab in the paediatric age group, and could comment as to whether this might influence treatment. Also it is worth commenting that none of these patients had surgically difficult to access sites eg vertebral column/sacrum etc

Author Response

Response to Reviewer 2 Comments

This is a well-written paper covering a very small patient population.

Point 1: Do the authors have any experience with the use of denosumab in the paediatric age group, and could comment as to whether this might influence treatment.

Response 1: Thank you for your comments. We did not use denosumab in the treatment of all patients. The reason for exclusion of denosumab in the treatment is added in lines 156-158.

Lines 156-158: The use of denosumab could be taken into consideration; however, we did not use it because sufficient resections were already performed in all cases. Besides, there were many concerns about adverse effects such as rebound bone turnover and hypercalcemia(14).

Point 2: Also it is worth commenting that none of these patients had surgically difficult to access sites eg vertebral column/sacrum etc

Response 2: We agree with your valuable suggestion. None of the patients had GCT in lesions that were difficult to access surgically, such as the vertebral column or sacrum. We have added these details in lines 154-158.

Lines 154-158: None of the patients had GCT in lesions that were difficult to access surgically, such as the vertebral column or sacrum. The use of denosumab could be taken into consideration; however, we did not use it because sufficient resections were already performed in all cases. Besides, there were many concerns about adverse effects such as rebound bone turnover and hypercalcemia(14).

Round 2

Reviewer 1 Report

The author corrected the manuscript appropriately following my suggestions. I have no further question.